# Whole-Exome Screening and Analysis of Signaling Pathways in Multiple Endocrine Neoplasia Type 1 Patients with Different Outcomes: Insights into Cellular Mechanisms and Possible Functional Implications

**DOI:** 10.3390/ijms25021065

**Published:** 2024-01-15

**Authors:** Anna Skalniak, Małgorzata Trofimiuk-Müldner, Marcin Surmiak, Justyna Totoń-Żurańska, Agata Jabrocka-Hybel, Alicja Hubalewska-Dydejczyk

**Affiliations:** 1Department of Internal Medicine, Jagiellonian University Medical College, 31-066 Krakow, Poland; anna.skalniak@uj.edu.pl; 2Department of Endocrinology, Jagiellonian University Medical College, 30-688 Krakow, Poland; malgorzata.trofimiuk@uj.edu.pl (M.T.-M.); agata.jabrocka-hybel@uj.edu.pl (A.J.-H.); alicja.hubalewska-dydejczyk@uj.edu.pl (A.H.-D.); 3Center for Medical Genomics—OMICRON, Jagiellonian University Medical College, 31-034 Krakow, Poland; justyna.toton-zuranska@uj.edu.pl

**Keywords:** multiple endocrine neoplasia type 1 (MEN1), whole-exome screening, pituitary tumors, adrenocortical tumors

## Abstract

Multiple endocrine neoplasia type 1 (MEN1) is a syndrome characterized by tumors in multiple organs. Although being a dominantly inherited monogenic disease, disease phenotypes are unpredictable and differ even among members of the same family. There is growing evidence for the role of modifier genes in the alteration of the course of this disease. However, genome-wide screening data are still lacking. In our study, we addressed the different outcomes of the disease, focusing on pituitary and adrenocortical tumors. By means of exome sequencing we identified the affected signaling pathways that segregated with those symptoms. Most significantly, we identified damaging alterations in numerous structural genes responsible for cell adhesion and migration. Additionally, in the case of pituitary tumors, genes related to neuronal function, survival, and morphogenesis were repeatedly identified, while in patients with adrenocortical tumors, *TLR10*, which is involved in the regulation of the innate immunity, was commonly modified. Our data show that using exome screening, it is possible to find signatures which correlate with the given clinical MEN1 outcomes, providing evidence that studies addressing modifier effects in MEN1 are reasonable.

## 1. Introduction

Multiple endocrine neoplasia type 1 (MEN1) is a monogenic, dominantly inherited disorder with a prevalence of approx. 1:20,000, caused by pathogenic variants in the gene *MEN1*. The outcome of the disease includes multiple endocrine and non-endocrine tumors. The disease is highly penetrant and will develop in all disease-causing allele carriers of the *MEN1* gene throughout their lives [1]. However, the course of the disease is unpredictable based on the underlaying *MEN1* mutation, and even within families, the grade of disease aggressiveness, the presence or absence of given symptoms, and their order of appearance and intensity differ strongly between individuals [2,3,4]. Due to the lack of clear correlations between the *MEN1* variant type or localization with the course of the disease, as well as the variation in phenotypes observed in carriers of the same disease-causing *MEN1* variant (e.g., in families), it has previously been implied that modifier genes might be responsible for masking or influencing the relationship between the *MEN1* variant and the clinical expression of the disease [5,6]. Based on animal studies, it has been suggested that the genomic landscape of the individual may be responsible for differences in the symptoms that occur in different MEN1 cases. *Men1*-knockout mice and heterozygous *Men1*^+/−^ mice on different backgrounds reveal clinical differences, including the risk and outcome of tumor development, which suggests a role for modifier genes [7,8,9]. The *MEN1*-encoded protein menin is known to interact with numerous proteins of different functions [1] and to be a transcriptional regulator for many genetic regions [10,11]. Therefore, the modifying impact on the disease introduced by mutations in other genes seems to be probable. However, until now, no attempts have been made to screen the genetic background in human MEN1 patients, which would allow for the identification of genetic modifiers that influence the predisposition to different disease symptoms and are not predefined by literature data.

We performed exome sequencing with subsequent filtering of genes carrying damaging variants. Gene ontology and pathway analyses were conducted to identify overrepresented biological processes in patients with a given disease outcome. We have recently published results for the pancreatic aspect of the disease (including insulinoma) [12]. Here, we analyze the state of pituitary tumors and adrenocortical tumors in the course of MEN1.

The aim of this study was to preliminarily verify the existence of differences in the genomes of MEN1 patients, that would associate with pituitary tumors or adrenocortical tumors. The present study also aimed at verifying whether there may be biological processes altered due to encoded changes in the patients’ genomes that would segregate with the different symptoms of the disease.

## 2. Results

### 2.1. Pituitary Tumors

None of the identified genes was common for all symptom-positive subcase groups. Pathways shared between the groups are listed in Figure 1 (for details including gene lists and statistical descriptions see Appendix A). A number of the shared pathways were identified due to the repeated presence of several genes in those cascades. The main functions of the involved genes were analyzed based on the descriptions in Entrez and UniProt. Genes that are involved in the shared pathways and that have repeatedly been identified in different patient subgroups functionally encompass the structural integrity of tissues; morphogenesis, including specifically the morphogenesis of the nervous system; neuronal function and survival; interstrand DNA cross-link repair; and glucose and energy metabolism.

### 2.2. Adrenocortical Tumors

In the four analyzed patient groups, one commonly mutated gene with a function-disturbing variant was identified in each of the groups—*TLR10*, which is addressed in the Section 3.

The following GO biological processes were overrepresented in all patient groups: anatomical structure development, cellular developmental process, organelle organization, cellular localization, ion homeostasis, chemical homeostasis, Toll-like receptor 10 signaling.

The common affected pathways are presented in Figure 2 (for details including gene lists and statistical descriptions see Appendix A).

## 3. Discussion

### 3.1. Genes and Pathways Identified in the Pituitary Tumor Patient Groups

Pituitary tumors can be either isolated or syndromic. In the second case, they can occur in different tumor predisposition syndromes. Besides *MEN1*, pathogenic variants in many other genes may predispose to the occurrence of these tumors, and depending on their genetic background, they will have different molecular characteristics. Also, differential expression of genes and epigenetic changes may influence an individual’s predisposition to those tumors as well as the tumor’s clinical appearance like tumor size, invasive behavior, or hormone secretion [13]. The amount and diversity of genomic alterations that may be involved in the clinical picture of pituitary tumors, together with the numerous interaction partners for the *MEN1*-encoded protein menin and the pathways it is involved in, may speak to a potential role of modifiers in the development and clinical picture of pituitary tumors in the course of MEN1.

In our study, analyses of differential gene variations between pituitary tumor-positive and pituitary tumor-negative MEN1 patients have shown the potential for dysregulation of functions and processes like the neuronal function, survival and morphogenesis of the nervous system, tissue integrity, and morphogenesis. Identified genes involved in those processes include the nerve growth factor *NGF*; *CHAT* which is involved in neurotransmitter biosynthesis; *NOTCH3*, which regulates cell-fate determination [14]; the neurotrophic factor *GPI*; *FRAS1*—a gene involved in brain organization and function [15]; *GFAP*—involved in neural crest development [16]; *RELN* and *TNC*, which regulate neuronal migration and the mobility of neuroblasts [17,18]; *KIF5C*, which is involved in synaptic transmission [19]; and others. This variety shows that alterations at different levels of the functioning of the nervous system may be involved in the development of pituitary tumors in MEN1 patients.

Only few of the pathways were represented statistically significantly (*p*-value < 0.05) in all patient subgroups. In the inflammatory response pathway (WP453), pathogenic variants, including at least one rare variant (with a frequency below 0.01) have been identified in each of the symptom-positive patient subgroups in genes encoding structural proteins responsible for cell adhesion and migration, including embryogenesis. In the PI3K-Akt signaling pathway (WP4172), several genes were detected to be altered in the pituitary tumor-positive patients, including rare variants in each of the subgroups. Again, the rare variants were found mainly in structural genes responsible for cell adhesion and migration but also in growth factors and growth factor receptors. The PI3K-Akt signaling pathway is known to be one of the most crucial pathways to regulate cell survival and metabolism and its differential regulation has been associated with multiple tumors. Many drugs targeting PI3K signaling have already been and are still being developed for cancer treatment [20]. The identification of this pathway in the context of the predisposition to tumors also in the course of familial syndromes seems sensible.

The identification of significantly altered structural genes, including different collagen subgroups, is in line with the fact that mechanical cell–cell interactions as well as interactions with the extracellular matrix are critical factors for tumor progression [21].

### 3.2. Genes and Pathways Identified in the Adrenocortical Tumor Patient Groups

In our MEN1 patients with adrenocortical tumors, among the cascades with altered genes, one small pathway—with 18 nodes—was identified, i.e., WP3877—the MYD88 distinct input–output pathway. This pathway transduces signals in the canonical activation of NF-kB and AP-1 in interleukin-1 and Toll-like receptor signaling, and also induces the production of reactive oxygen species. In all patient groups from our study, this pathway was identified due to an alteration in the Toll-like receptor gene *TLR10*. Toll-like receptors play a fundamental role in the activation of innate immunity. While the variant rs11096957 (p.Asn241His) in the *TLR10* gene identified in our study is common, with an allele frequency of 0.3691 in the European, non-Finnish population, it was absent in all controls while present in all MEN1 patients with adrenocortical tumor in either a heterozygous or homozygous state. SIFT prediction, which assesses whether a variant affects protein function, based on the sequence homology and physical properties of the encoded amino acids, scored this variant as “damaging”. Polyphen, which predicts how a substitution may impact the structure and function of an encoded protein, classified this variant as “possibly damaging”. According to FATHMM, a tool that predicts the functional consequences of variants based on hidden Markov models, also classified the variant as “damaging”, and mutation assessor, which analyses a variant’s functional impact on the protein based on evolutionary conservation of the amino acid in homologs of the protein, classified the impact risk as “medium”, i.e., having a predicted functional impact. In contrast, the multi-variate genetic variant classification according to ACMG, which predicts the pathogenicity of single variants with respect to clinical phenotypes, classifies this variant as benign. Also, MutationTaster, a tool that identifies the disease-causing potential of genetic variants, does not classify it as disease-causing. This means that according to prediction tools, this variant, while not being itself responsible for a specific clinical condition in the variant-carrier, is predicted to have an impact on the encoded protein’s function; therefore, it seems to be a good candidate for a modifying variant, as the encoded protein, with an altered function, might interact with other structures to alter the faith of a cell or tissue, if present in predisposing conditions.

A number of pathways, i.e., WP2118-arrhythmogenic right ventricular cardiomyopathy, WP2572-primary focal segmental glomerulosclerosis, WP2911-miRNA targets in ECM and membrane receptors, WP383-striated muscle contraction, WP4352-ciliary landscape, WP4754-IL-18 signaling pathway, were identified due to variants in structural protein-encoding genes. In addition to them, developmental pathways were identified, including the neural crest differentiation pathway, the ectoderm differentiation pathway, the TGFb signaling pathway, and a Notch3-associated pathway. Cascades related to oxidative stress included the NRF2 pathway, the VEGFA-VEGFR2 signaling pathway, the MYD88 pathway, and the selenium micronutrient network.

The only networks that were statistically significant (*p*-value < 0.05) in each of the analyzed subgroups, consisted of focal adhesion networks, WP3932 (including PI3K-Akt-mTOR signaling) and WP306, and the PI3K-Akt signaling pathway WP4172. In those pathways, each patient group had 3–5 genes with damaging variants, that are involved in extracellular matrix (ECM)–integrin receptor interactions: patient group A—*ITGAE*, *ITGB4*, *LAMA4*, *THBS4*; patient group C—*COL11A2*, *COL4A4*, *COL5A1*, *COL6A2*, *TNC*; patient group X11—*COL5A1*, *LAMA5*, *RELN*; patient group X15—*COL4A4*, *FN1*, *LAMA1*, *RELN*, *TNC*. Cell-matrix adhesion is crucial in biological processes such as embryonic development, cell proliferation, differentiation and survival, cell motility, and tissue homeostasis [22].

### 3.3. Implications for the Future

In our study, we analyzed the genetic background of MEN1 patients. The inclusion of family members in the analyses allowed us to exclude a larger proportion of the insignificant genetic background than would be the case with unrelated study participants, while including unrelated patients led to the identification of unique patterns in the genome that are found in a broader population than a single family.

The presented analyses are of preliminary nature and have been undertaken due to the lack of genome screening analyses in MEN1 patients with different disease outcomes and the limited knowledge on the role of genetic modifiers in the occurrence of specific symptoms in MEN1-affected individuals. The patient group was limited most significantly due to the rarity of the disease and the need to include families with specific combinations of clinical outcomes but of similar ages.

It must be noted that we do not infer that the identified pathways or functional groups of genes are the only ones to be considered in further discussions on modifying effects in MEN1. The *MEN1*-encoded gene menin is known to interact with numerous proteins to fulfil its functions. It is involved in the regulation of gene expression in different ways. Those include direct interactions with multiple transcription factors, like JunD, NF-κB, Myc, or VDR, among others. Also, its interactions with histone-modifying enzymes and the polycomb group and thus its modification of chromatin influences gene transcription by changing the accessibility of gene promoters to transcriptional factors. Furthermore, menin can also act as a transcription factor by directly interacting with gene promoters. Menin has also previously been shown to interfere with different cell signaling pathways, including TGF-β and Wnt signaling, and downstream targets of the PI3K/AKT pathway alter menin expression [1,23,24]. It is therefore probable that variations in different pathways and genes, including those identified in our study, may lead to disease-altering predispositions among MEN1 patients. Different modifiers and combinations of modifiers may lead to similar disease outcomes, and the results we obtained most probably encompass only some of the possibilities.

However, our data are the first to implement studies on the modifying effects of genetic background on MEN1 outcomes in human patients, without being bound by literature-predefined genes included in the analysis. Our investigation is meant to indicate an exemplary direction in which much work is left to be completed in order to attain a more comprehensive understanding of the complexity of MEN1. Further insight into genetic modifiers may enable the prediction of the course of the disease and allow for elucidating the networks involved in its pathogenesis. Importantly, dependent on the trait, genetic modifiers can even suppress monogenic and multigenic traits in otherwise susceptible individuals [25,26].

Further studies on modifying effects in MEN1 might encompass broader analyses on genetic background, including gene expression and epigenetic control. It is known that MEN1 undergoes different posttranslational modifications that modify its function; however, the functional impact of those modifications has not yet been investigated [1]. Studies addressing the role of miRNAs in MEN1 are being undertaken. For an exemplary review on this topic, see Donati et al. [27]. Large-scale screening studies to identify miRNAs that are altered in MEN1 patients have also been undertaken lately. The identified miRNAs might be altered as an effect of tumorigenesis but could also be addressed in future studies to verify if they may influence the transition of normal cells into tumor cells or the transition into given tumor types [28,29,30]. Last but not least, environmental factors should also be considered as possible significant factors. Altogether, the whole picture of the disease might be best described by a complex, multiparametric model. However, until now, studies that address the problem of disease outcome modifiers in human MEN1 patients have focused on predefined ideas, where the investigated factor has been included in the study based on previous knowledge. According to the best of our knowledge, our study is the first to address MEN1 modifiers that allowed for the large-scale detection of genetic factors without being restrained to factors predefined by the literature.

## 4. Methods

### 4.1. Patients

Fifteen symptomatic patients with genetically confirmed MEN1 were invited to participate in the study, including members from 4 families. In total, 10 of the patients, including 6 members from 2 families, were classified for downstream analyses based on their clinical characteristics, the remainder were included in different analyses (published elsewhere, [12]). The pathogenicity of the germline *MEN1* variants was verified based on their deleterious impact on the gene structure (frameshift and nonsense variants, large deletions) and available databases (Varsome [31], NCBI ClinVar [32], LOVD [33]). Only in family A, a previously unreported in-frame variant was identified. We have published the proof of pathogenicity of this variant in a different paper, which encompassed co-segregation analysis and identification of the second hit in the tissue of an affected family member [34].

Two separate analyses were performed in the present study: the presence vs. absence of (1) a pituitary tumor, and (2) an adrenocortical tumor in the course of MEN1. All patients except one were over 40 years old at the time of the analyses (Table 1), as this is the age by which all clinical manifestations of MEN1 are already present in 95% of patients [1]. Therefore, the age below 40 years was an exclusion criterion for the control group (symptom-negative) in each of the analyses. The only younger patient (C9, 33 years) was included only in the adrenocortical tumor analysis, in the positive group of patients.

The analyses were performed in patient subgroups; in each of them, the symptom-positive patients included all positive cases of a given family or one patient who was unrelated to any other study participant. The symptom-negative group included unrelated patients as well as a maximum of one family member in the case of family analyses.

In the pituitary tumor analysis, the patients A2, X11, and X12 were included as symptom-negative controls. The pituitary tumor-positive patient subgroups were family A (combined analysis of positive A4 and A8), patient X5, and patient X15.

In the adrenocortical tumor analysis, symptom-negative controls included, in all analyses, X5, A8, X10, and X12, while the symptom-positive subgroups were family A (patient A4), family C (C3 and C9), and patients X11 and X15. No family member in family C met the criteria of a negative control; however, patient X10 bore the same disease-causing *MEN1* variant as family C, despite being unrelated to this family at least 2 generations back, as was determined based on interviews with both families. This patient was included as negative control in the adrenocortical tumor analysis and served as background for family C to ensure the exclusion of any potential modifier effects that might result from the disease-causing *MEN1* variant.

### 4.2. Ethical Statement

The study was conducted in accordance with the principles set out in the Declaration of Helsinki and the study design was approved by the Bioethics Committee of the Jagiellonian University in Krakow, Poland (Opinion No. 122.6120.267.2015). The study participants gave their informed consent for genetic analyses within the scope of the study.

### 4.3. Genomic DNA Extraction and Whole Exome Library Preparation

Blood was collected from each participant into anticoagulation tubes and genomic DNA was isolated on the Maxwell^®^ 16 Instrument, using the Maxwell^®^ 16 LEV Blood DNA Kit (Promega, Madison, WI, USA).

Quantification of the DNA was performed spectrophotometrically on NanoDrop and fluometrically on Quantus (Promega), using the QuantiFluor dsDNA system. After shedding in the Bioruptor instrument (Diagenode, Denville, NJ, USA), sample quality was assessed with the Agilent High Sensitivity D1000 ScreenTape assay on TapeStation (Agilent, Santa Clara, CA, USA). The Agilent Technologies SureSelect XT Reagent Kit was used to obtain whole-exome sequencing libraries, and exones were captured with the OneSeq Constitutional Research Panel (Agilent). The libraries were indexed in a 10-cycle PCR reaction and samples were multiplexed in equal molar concentrations and subsequently sequenced on a HiSeq sequencing device (Illumina, San Diego, CA, USA) by an external service provider (EMBLEM, Heidelberg, Germany).

### 4.4. High Throughput Data Analysis

Base calls and base-call quality scores were obtained from the reads by means of the device-associated Illumina software (1.8). FastQC (v. 0.11.5) was used to perform quality control checks on fastq files [35]. The human reference genome GRCh37 (hg19) was used for alignment with the BWA-MEM algorithm in BWA (Burrows Wheeler Aligner, v.0.7.5) [36]. Unmapped reads and reads with low mapping quality scores were filtered out with SAMtools (v. 4.0) (Strand Life Sciences Pvt. Ltd., Bangalore, India), which was used also for further downstream analyses. Variants were called based on dbSNP151 indicators. The variant lists were filtered for significant variants, i.e., alleles absent in all negative controls but present in all symptom-positive patients, or alleles that were homozygous in symptom-positive patients while being in a heterozygous state in negative controls. The obtained variants were filtered to receive a list of “damaging” variants only, which are defined as not neutral to the encoded gene product, as indicated by at least 3 tools from among: SIFT [37], LRT [38], MutationTaster [39], Polyphen2 (HumDiv or HumVar) [40], MutationAssessor [41], FATHMM [42], and the meta predictors MetaSVM and/or MetalR [43], and which do not occur in the same zygosity state in symptom-positive and symptom-negative cases in a given analysis. Genes with the identified variants were listed. Following this procedure, we obtained one list for each subgroup of families or unrelated cases used in the analyses, with genes differing in non-neutral variants between symptom-positive and symptom-negative patients.

### 4.5. Downstream Analyses

In silico analyses (pathway detection and gene ontology (GO)) revealed functional enrichment of the mutated genes. Gene ontology (GO) analyses were performed with the online tool Panther17.0 [44] (resource release 22 March 2022). Overrepresented biological processes were identified using Fisher’s exact test with a false discovery rate (FDR) calculation. Minor allele frequencies were obtained from gnomAD version 2.1.1 for the European, non-Finnish population [45]. Pathway analyses were performed in Strand NGS (WikiPathways version 20210910).

## 5. Conclusions

Our study is the first to address the different outcomes of MEN1 in means of pituitary tumor and adrenocortical tumor development based on genome-wide searching, without being restricted by previous knowledge-based choices of the analyzed genes. The screening of the genetic background of MEN1 patients revealed in both analyses a relationship with structural genes responsible for cell adhesion and migration. In the pituitary patient group, there were repeatedly genes identified that are involved in the development, integrity and functioning of the nervous system. In all patient subgroups of the adrenal analysis, alterations were found in a gene with an impact on the activation of proinflammatory signals in innate immunity. Most significantly, our study provides evidence that studies addressing genetic modifier effects in MEN1 are reasonable and might be planned in a broader range in the future.

## Figures and Tables

**Figure 1 ijms-25-01065-f001:**
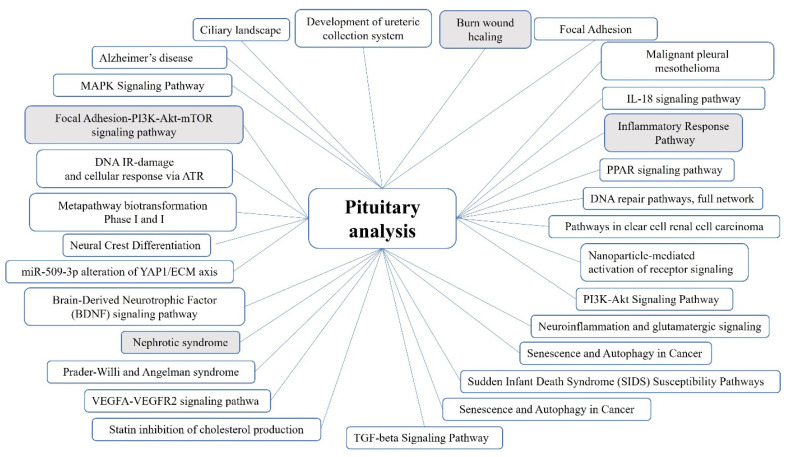
Pathways shared between all patient groups in the MEN1 pituitary analysis. Pathways that are statistically significant in all family and patient groups are shaded.

**Figure 2 ijms-25-01065-f002:**
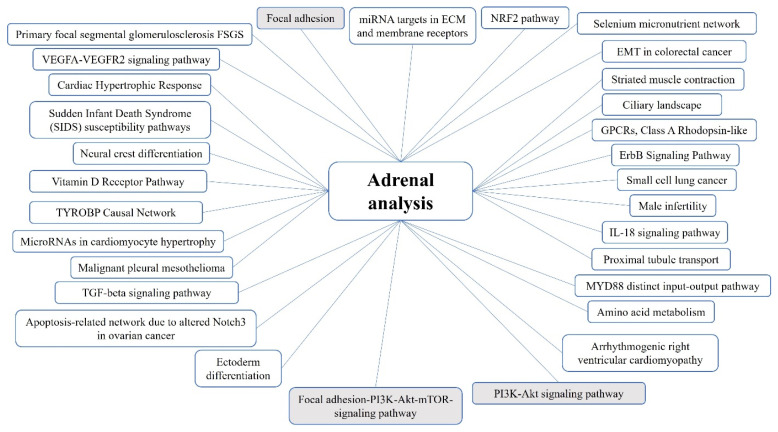
Pathways shared between all patient groups in the MEN1 adrenal analysis. Pathways that are statistically significant in all family and patient groups are shaded.

**Table 1 ijms-25-01065-t001:** Patients included in the analyses.

Patient ID	Family ID *	Disease-Causing *MEN1* Variant	Age at Time of Analysis	Sex	Primary Hyperpara-Thyroidism	Pituitary Tumor	Neuro-Endocrine Neoplasm (NEN)	Pancreatic NEN	Adrenocortical Tumor	Other	Included in Pituitary Analysis (Patient or Group ID)	Included in Adrenal Analysis
A2	A	c.1246_1248delGCC/p.Ala416del	59	F	yes	No	no	yes	no	n/d	neg	-
C3	C	c.945delG/p.Tyr316Profs*57	59	M	yes	No	yes	yes	yes	n/d	-	pos(C)
A4	A	c.1246_1248delGCC/p.Ala416del	54	M	yes	Yes	no	yes	yes	n/d	pos(A)	pos(A)
X5	X	c.866C>A/p.Ala289Glu	59	F	yes	Yes	no	no	no	nasal cavity cancer	pos(X5)	neg
A8	A	c.1246_1248delGCC/p.Ala416del	48	M	yes	yes	no	no	no	n/d	pos(A)	neg
C9	C	c.945delG/p.Tyr316Profs*57	33	M	yes	no	no	no	yes	adrenocortical oncocytoma	-	pos(C)
X10	X	c.945delG/p.Tyr316Profs*57	72	F	yes	n/d	yes	yes (insulinoma)	no	n/d	-	neg
X11	X	c.416A>G/p.His139Arg	75	F	yes	no	yes	yes (insulinoma)	yes	n/d	neg	pos(X11)
X12	X	c.796C>T/p.Gln266Ter	65	F	yes	no	yes	yes	no	n/d	neg	neg
X15	X	c.1256delTGC/p.Leu419del	75	M	yes	yes	yes	no	yes	n/d	pos(X15)	pos(X15)

* A, C—family identifiers; X—unrelated to any of the other study participants; F, female; M, male; n/d, no data; neg, symptom-negative; pos(), symptom-positive (identifier of the analyzed patient or patient group); “-”, not included in this analysis.

## Data Availability

Data will be made available on request.

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
