# Peer review of "Whole-Exome Screening and Analysis of Signaling Pathways in Multiple Endocrine Neoplasia Type 1 Patients with Different Outcomes: Insights into Cellular Mechanisms and Possible Functional Implications"

_ijms, 2024, doi:10.3390/ijms25021065_

Round 1
Reviewer 1 Report
Comments and Suggestions for Authors
This article presented here provides novel insights into genes and pathways that could correlate with MEN1 Syndrome.
This article is fascinating and scientifically sound. I have only a few considerations that could further improve the quality of the manuscript, which are listed below:
1. it could be interesting, whether you have data available for the homo-/heterozygous state in MEN1 tumors from your study (this latter information could be added in the main text and Table 1), to identify common genes that could distinguish the homozygous state from that heterozygous one;
2. considering that Menin has been demonstrated to be primarily a nuclear protein, are there data in the literature about the possible ability of Menin to regulate the expression of genes identified in your analysis? In that case, you could discuss it in the discussion section.
Finally, as you state in the discussion section, epigenetic factors, including microRNAs, could play a crucial role in this syndrome. It would be exciting to provide new insights about their involvement in MEN1 by correlating with the pathways identified in this study. In this regard, just as a matter of curiosity, are you planning to investigate these factors?
All in all, I congratulate the authors for their hard work, and I look forward to seeing this paper published in the journal.
Author Response
Point 1: “It could be interesting, whether you have data available for the homo-/heterozygous state in MEN1 tumors from your study (this latter information could be added in the main text and Table 1), to identify common genes that could distinguish the homozygous state from that heterozygous one.”
Response 1: While indeed, it has been shown that a heterozygous microenvironment might accelerate benign tumor formation in different hereditary cancer syndromes, in the case of MEN1, the mechanism of disease development is based on the biallelic inactivation of MEN1. The first, heterozygous loss-of-function variant in the gene is inherited by the patient (homozygous germline loss of MEN1 alleles is lethal to the foetus). For the development of the tumor, the two-hit mechanism consistent with Knudson’s theory is required. This second hit, i.e. the somatic loss-of-function variant in the second allele, is acquired during the lifetime of the patient and results in the tumorigenesis of the affected tissue and will occur in all affected tissues, although by different mechanisms (large chromosomal deletions as well as small variants) – see for example https://pubmed.ncbi.nlm.nih.gov/11549677/, https://www.ncbi.nlm.nih.gov/pmc/articles/PMC7958143/.
Therefore, the zygosity state is not tested in MEN1 tumors, unless needed for the confirmation of the pathogenic character of a variant. The inherited pathogenic variants in MEN1 are family-specific and no unequivocal genotype-phenotype correlations have been described despite multiple attempts to do so. However, many of the so-far identified variants have been deposited in databases, together with information regarding their pathogenicity. Therefore, only newly identified variants in MEN1 require individual attempts to prove their pathogenicity. In the clinical practice, variants which are definitely deleterious to the structure of the gene like large deletions or mutations which introduce a premature stop codon in the menin protein-coding region of the MEN1 gene are assessed only bioinformatically, and tissues are not usually tested. In the case of any other variant, it has to be defined whether it is a pathogenic or a neutral change. In our patients, only family A had a newly identified variant which needed to be verified. This verification has been published in our previous paper, which we have now added to the manuscript. In this paper, besides co-segregation of the variant with affected family members, we have also proved the loss of the healthy allele in the patient’s tissue. All other MEN1 cases are assumed to have homozygous states in their tissues, based on previous knowledge.
Point 2: “Considering that Menin has been demonstrated to be primarily a nuclear protein, are there data in the literature about the possible ability of Menin to regulate the expression of genes identified in your analysis? In that case, you could discuss it in the discussion section.”
Response 2: Menin is known to be directly interacting with multiple transcription factors including JunD, NF-kB, Myc, PPARg, VDR, and many more. In addition, it interacts with histone-modifying enzymes and the polycomb group, so that it influences gene transcription through modification of chromatin and thus the accessibility of gene promoters for transcriptional factors. Menin can also directly interact with gene promoters as a transcription factor. (For references, see https://www.ncbi.nlm.nih.gov/pmc/articles/PMC7958143/, https://www.ncbi.nlm.nih.gov/pmc/articles/PMC5966343/, https://www.ncbi.nlm.nih.gov/pmc/articles/PMC5612327/.) Due to the multiple ways of interacting with gene expression, the discussion of all the genes identified in our study that are directly or indirectly regulated by menin, would result in a volume of analyses, which would exceed the feasibility of including them in this brief communication-type publication. However, we have added this issue to the discussion, as indeed, this question might arise also in future readers of the paper.
Point 3: Finally, as you state in the discussion section, epigenetic factors, including microRNAs, could play a crucial role in this syndrome. It would be exciting to provide new insights about their involvement in MEN1 by correlating with the pathways identified in this study. In this regard, just as a matter of curiosity, are you planning to investigate these factors?
Response 3: There are a number of studies on microRNAs ongoing with some of them already published by different teams of investigators, including researchers who have also previously made a very significant contribution to the development of knowledge in the field of MEN1. In those studies, microRNAs that are differently regulated in tissues or in the blood of subjects with MEN1-associated tumors are investigated. For examples, see https://pubmed.ncbi.nlm.nih.gov/34298972/, https://pubmed.ncbi.nlm.nih.gov/35900839/, https://pubmed.ncbi.nlm.nih.gov/34285285/, https://www.ncbi.nlm.nih.gov/pmc/articles/PMC8023566/, https://www.ncbi.nlm.nih.gov/pmc/articles/PMC7589704/.
It would indeed be interesting to correlate those data with our results. However, in this case, it would be necessary to analyse microRNAs and genomic data in the same patients, because changes in microRNA expression represent also changes that have appeared as a consequence of tumorigenesis and differ between patients, even more importantly if we wish to detect associations between identified pathways and microRNAs in individual subjects. We currently do not have any ongoing projects that would address this issue and have not planned to request funding for this kind of project.
We have, however, added exemplary citations to our manuscript, for other readers’ information in case of their curiosity about this topic. Thank you for addressing this issue.
Reviewer 2 Report
Comments and Suggestions for Authors
The authors present a way of attempting to identify modifiers that can be implicated in explaining the phenotypic variation that is known to be prevalent in MEN. While no new modifiers, gene variants or pathways have been irrefutably identified, the methodology does provide some direction. Further analysis, as well as functional studies would be needed to consolidate this approach.
Author Response
Thank you for your time spent to read the manuscript and for your comments. Indeed, we hope that our study will provide some direction for future research.